# Interlaboratory Comparison Study on Ribodepleted Total RNA High-Throughput Sequencing for Plant Virus Diagnostics and Bioinformatic Competence

**DOI:** 10.3390/pathogens10091174

**Published:** 2021-09-12

**Authors:** Yahya Z. A. Gaafar, Marcel Westenberg, Marleen Botermans, Krizbai László, Kris De Jonghe, Yoika Foucart, Luca Ferretti, Denis Kutnjak, Anja Pecman, Nataša Mehle, Jan Kreuze, Giovanna Muller, Nikolaos Vakirlis, Despoina Beris, Christina Varveri, Heiko Ziebell

**Affiliations:** 1Institute for Epidemiology and Pathogen Diagnostics, Julius Kühn Institute (JKI)–Federal Research Centre for Cultivated Plants, Messeweg 11/12, 38104 Braunschweig, Germany; yahya.gaafar@julius-kuehn.de; 2National Reference Centre of Plant Health, Dutch National Plant Protection Organization, Geertjesweg 15, 6706 EA Wageningen, The Netherlands; m.westenberg@nvwa.nl (M.W.); m.botermans@nvwa.nl (M.B.); 3Plant Health Diagnostics National Reference Laboratory, Directorate of Food Chain Safety Laboratory, National Food Chain Safety Office, Budaörsi út 141–145, H-1118 Budapest, Hungary; KrizbaiL@nebih.gov.hu; 4Plant Sciences Unit, Flanders Research Institute for Agriculture, Fisheries and Food (ILVO), Burgemeester Van Gansberghelaan 96, 9820 Merelbeke, Belgium; kris.dejonghe@ilvo.vlaanderen.be (K.D.J.); Yoika.foucart@ilvo.vlaanderen.be (Y.F.); 5Research Centre for Plant Protection and Certification, Council for Agricultural Research and Economics, Via C.G. Bertero 22, 00156 Rome, Italy; luca.ferretti@crea.gov.it; 6Department of Biotechnology and Systems Biology, National Institute of Biology, Večna pot 111, SI-1000 Ljubljana, Slovenia; denis.kutnjak@nib.si (D.K.); anja.pecman@nib.si (A.P.); natasa.mehle@nib.si (N.M.); 7Jožef Stefan International Postgraduate School, SI-1000 Ljubljana, Slovenia; 8Health and Quarantine Unit, International Potato Center (CIP), Av. La Molina 1895 La Molina, Lima 15023, Peru; J.KREUZE@CGIAR.ORG (J.K.); g.muller@cgiar.org (G.M.); 9Benaki Phytopathological Institute, Stefanou Delta 8, Kifissia, Attica, 14561 Athens, Greece; vakirlis@fleming.gr (N.V.); d.mperi@bpi.gr (D.B.); c.varveri@bpi.gr (C.V.)

**Keywords:** high-throughput sequencing, ribodepletion, interlaboratory comparison, test performance study, proficiency test, Virtool

## Abstract

High-throughput sequencing (HTS) technologies and bioinformatic analyses are of growing interest to be used as a routine diagnostic tool in the field of plant viruses. The reliability of HTS workflows from sample preparation to data analysis and results interpretation for plant virus detection and identification must be evaluated (verified and validated) to approve this tool for diagnostics. Many different extraction methods, library preparation protocols, and sequence and bioinformatic pipelines are available for virus sequence detection. To assess the performance of plant virology diagnostic laboratories in using the HTS of ribosomal RNA depleted total RNA (ribodepleted totRNA) as a diagnostic tool, we carried out an interlaboratory comparison study in which eight participants were required to use the same samples, (RNA) extraction kit, ribosomal RNA depletion kit, and commercial sequencing provider, but also their own bioinformatics pipeline, for analysis. The accuracy of virus detection ranged from 65% to 100%. The false-positive detection rate was very low and was related to the misinterpretation of results as well as to possible cross-contaminations in the lab or sequencing provider. The bioinformatic pipeline used by each laboratory influenced the correct detection of the viruses of this study. The main difficulty was the detection of a novel virus as its sequence was not available in a publicly accessible database at the time. The raw data were reanalysed using Virtool to assess its ability for virus detection. All virus sequences were detected using Virtool in the different pools. This study revealed that the ribodepletion target enrichment for sample preparation is a reliable approach for the detection of plant viruses with different genomes. A significant level of virology expertise is needed to correctly interpret the results. It is also important to improve and complete the reference data.

## 1. Introduction

Plant viruses and virus-like diseases are ever-emerging threats to agricultural and horticultural production [1]. To ensure that plants are free from regulated viruses, a reliable and rapid method is required for the accurate identification of the pest. Conventional detection methods for plant viruses and virus-like diseases including molecular, serological, and biological indexing are primary tools used by plant virologists [2,3]. Conventional methods rely on previous information of the virus, e.g., the virus sequence, to design primers for PCR-based methods [2] or purified virions for the production of virus-specific antibodies. The advances in high-throughput sequencing (HTS) technologies increased the breadth of plant viruses’ detection [4]. HTS does not require prior information as it enables the sequencing of all nucleic acids in a given sample [5]. However, viral nucleic acids require an enrichment method prior to the library construction for HTS as their sequences are present in the background of their host sequences [6]. There are different enrichment methods for virus sequences derived from plant samples, e.g., ribosomal RNA (rRNA) depleted total RNA (ribodepleted totRNA), double stranded (ds)RNA, and small (s)RNA isolations [6].

Total RNA (totRNA) was used as input for HTS. However, a potential limitation of this approach can be the low amount of viral RNA within the background of plant RNA. Virus reads generated by totRNA sequencing from plant material might be low in number and difficult to analyse due to the high amount of plant rRNA in the total RNA [5]. Ribodepleted totRNA methods with reduced ribosomal RNA from total RNA can enrich viral RNAs by ten-folds [6]. Thus, it is an efficient enrichment method for polyadenylated and non-polyadenylated RNA, pre-processed RNA, transfer RNA, regulatory RNA, and virus RNA. In comparisons between sRNA and ribodepleted totRNA, the latter was suggested to perform better for the detection of single-stranded RNA viruses [7].

To apply HTS as a routine tool for plant viruses’ detection, there are technical and biological challenges [8]. It is important to validate an approach to minimise the risk of false-negative and false-positive results [9]. A technical challenge for plant virus detection by HTS lies in the validation of the technology for the robust detection of a broad range of viruses and in determining the comparability of different approaches for acceptance in routine screening [8]. A critical challenge for the validation of HTS for plant virus detection is the bioinformatic analysis [8]. Bioinformatic analysis represents a core element in HTS. Expertise in bioinformatics as well as in virus taxonomy and diagnostics are required for the correct identification of the disease aetiology using HTS [10].

So far, no guidelines or standards on how to use HTS in a diagnostic setting are available. We therefore decided within the framework of the Euphresco project “The application of Next-Generation Sequencing technology for the detection and diagnosis of non-culturable organism: Viruses and viroids (NGS-detect)” to evaluate one viral enrichment method for its suitability for viral diagnostics. In this interlaboratory comparison, plant leaf material infected with viruses representing different genomic organisations, i.e., single stranded DNA and RNA (positive and negative single-stranded), was analysed using HTS of ribodepleted totRNA. The design of this part of the study had the characteristics of a test performance study (TPS) as all participants were required to follow the same protocol and use the same commercial sequencing provider. Participating laboratories were also assessed in terms of their capacity to analyse the sequencing data using their own bioinformatic pipeline, which has the characteristic of a proficiency test (PT). In addition, the data generated by each laboratory were analysed using Virtool to assess its performance.

## 2. Results

### 2.1. Invitation and Participation of Laboratories

Twenty laboratories were invited to participate in the study and nine laboratories agreed to participate in this study, of which eight performed the study and reported their results.

All laboratories were able to extract sufficient RNA for further processing (Table 1). None of the pooled samples were unsuitable for ribodepletion or HTS. The quantities of the RNA after ribodepletion varied between laboratories (Table 1). They ranged between 3.8 and 57.2 ng/µL for Pool 1, and between 4.2 and 43.7 ng/µL for Pool 2 (excluding TPS_07 because of unusual deviation). The number of raw reads obtained after sequencing also varied where the sequencing company provided between ≈ 13 and 54 M reads for Pool 1 (≈76 M for Lab ID TPS_05 where they used a different sequencing provider), and between ≈10 and ≈49 M reads (≈80 M for TPS_05) for Pool 2 (Table 1). There was no correlation between RNA quantity and the number of generated raw reads (data not shown). The quality of the generated raw reads by the providing company was close to each other with the average Q ≥ 30 of 92.5% (±0.3 SEM) for Pool 1 and 92.4% (±0.4) for Pool 2. The raw data generated from this study were deposited in the Sequence Read Archive (SRA) under BioProject accession number PRJNA737064, BioSample accession numbers SAMN19678996 to SAMN19679011, and SRA accession numbers SRR14794340 to SRR14794355.

### 2.2. In-House Bioinformatic Analysis by Participating Laboratories

Each laboratory used their own bioinformatic pipeline (Table 2), except for two laboratories that outsourced the analysis, i.e., TPS_01 and TPS_02. Four out of eight participants reported 100% accurate results for Pool 1, whereas three participants reported 100% accurate results for Pool 2 (Table 3). Two out of eight participants had 100% correct virus identification using the same bioinformatic pipeline, i.e., TPS_01 and TPS_08 (Table 2). Five participants failed to identify a novel virus at the time of the study, i.e., alfalfa associated nucleorhabdovirus (AaNV), in Pool 2 (Table 3). Laboratory TPS_02 reported false-positive results, i.e., avian coronavirus (AvCoV) and the citrus exocortis Yucatan viroid (a wrongly annotated sequence). Six viruses out of the ten included in Pool 1 were correctly identified by all laboratories, namely ACMV, AV-1, CGMMV, CLRV, CMoMV, and LNYV. In contrast, in Pool 2, only two viruses were identified correctly by all laboratories, i.e., CMV and ToMV. The mapped read percentages for each virus by the different laboratories were also different between datasets (Appendix A).

At a later step, the raw data were reanalysed using Geneious to generate virus consensus sequences. Consensus sequences generated from TPS_08 were used as references to map the reads from other participants, followed by alignment to confirm their similarity. BLASTn search revealed that ACMV, AaNV, CGMMV, CMoMV, CMV, PEMV2, PEMVsatRNA, PepMV, PhCMoV, and ToMV genomic sequences share high similarity to GenBank sequences, between 95.7 and 99.9% nt identity (Table 4). In contrast, AV-1, CLRV, ClYVV, CNDV, and LNYV are highly divergent sequences from the published sequences on GenBank, sharing only between 76.9 and 90.4% nt identity (Table 4).

### 2.3. Virtool Analysis

The datasets provided by the participant laboratories were reanalysed using Virtool. Twelve out of the fourteen viruses included in this study were detected by the PathoscopeBowtie approach. The percentages of mapped reads to the organisation taxonomic units (OTU) database of plant viruses were between 4.9 and 18.6% for Pool 1, and between 15.2 and 31.3% for Pool 2 (Figure 1). Both AaNV and PhCMoV were not in the OTU database of Virtool. To detect AaNV and PhCMoV, the NuVs approach was used and the sequences were detected in the Pool 2 data provided by each laboratory after the BLAST search tool in Virtool.

In order to compare the datasets, the raw reads were normalised to subsets of two million reads randomly extracted. Three replicates of the normalised reads were then reanalysed using Virtool. The calculated proportion of reads mapping to each virus (weight) are shown in Figure 2. The figure shows that the weights were slightly different from one laboratory to another. As expected, the weight of each virus was dependent on the virus and the “available” OTUs in the Virtool database. Four viruses have higher weights than the others, i.e., CGMMV, CMV, PepMV, and ToMV. Some of the plant virus OTUs in this version of Virtool were distantly related to the viruses in this study, e.g., AV-1, CNDV, CLRV, ClYVV, and LNYV, and are highly divergent isolates of the viruses (Table 4). Due to the weak relationship between these viruses and the OTUs in the Virtool database, a lower weight was observed for such cases (Figure 2). A similar scenario was observed for PEMV2 and its associated satellite PEMVSat; the available OTU is the reference sequence from GenBank, i.e., NC_003853 and NC_003854.

## 3. Discussion

High-throughput RNA sequencing is a robust virus diagnostic tool for plant virus identification and whole genome sequencing without prior knowledge of the pathogen(s) in the infected plant. In order to adopt it in routine plant virus diagnostics, this study was designed to assess:(a)the practical performance of each laboratory to prepare the samples for HTS;(b)their ability to correctly identify the viruses using their own bioinformatic strategies;(c)the checking of the variation in sequencing quality of one commercial provider; and(d)the performance of Virtool for data analysis as an alternative standard analysis option.

We opted to provide the infected material in the form of freeze-dried leaf material as, in general, freeze-dried material is more stable in transportation in contrast to fresh material. We expected delays in the delivery of the material, in particular to non-European countries, which proved correct in one case. Even in this case, all virus sequences could be recovered successfully. The participants were provided with individual vials of samples that had to be pooled by the participants themselves before extraction, as it was virtually impossible to harvest all viruses at the same time considering the infection dynamics were very different for the individual viruses (ranging from a few days to a few weeks for symptom development).

Host-sequence extraction using bioinformatic means is also a possible option to analyse HTS data. However, we opted for the sequencing of ribodepleted totRNA because host-sequence extraction requires host reference sequences which are not always available and higher bioinformatics power/infrastructure would have been necessary. Furthermore, the number of virus reads generated from totRNA HTS can be low due to the high background of host RNAs and can therefore lead to false-negative results. Increasing the sequencing depth could overcome these obstacles, although this comes at a higher cost. Ribodepleted totRNA is a suitable alternative approach that reduces the amount of extracted host and host-associated microbial rRNAs, thus enhancing the threshold of virus detection [6]. Despite the differences in the total amounts of extracted totRNA and ribodepleted totRNA obtained by the laboratories, and therefore the potential differences in the obtained sequence reads, these extracts were suitable for libraries preparation and HTS did not influence the interpretation of data and detection of viruses that were used in this study. In a different study carried out in the frame of the EUPHRESCO project NGS-detect, it was found that the amount of sequence reads had an influence on the false-negative or false-positive interpretation of HTS data (M. Rott, pers. comm.).

In this study, we were able to successfully detect ssDNA and ssRNA plant viruses. Other studies also showed that it is possible to detect dsDNA and dsRNA plant viruses using a ribodepleted totRNA approach [11]. This suggests that rRNA depletion is a robust method to provide RNAs for HTS analyses. Additionally, the library preparation by commercial suppliers and the quality of sequence reads generated by Illumina HiSeq 4000/NextSeq 500 machines were high and good for bioinformatic analysis. Alternative enrichment methods for viral sequences include the extraction of dsRNA [12]. dsRNA is a replicative form for many plant viruses and has been used for the enrichment of viral sequences to generate sequences from unknown plant viruses including RNA and DNA viruses.

Appropriate analysis of the data is a bottleneck for disease causal agent identification [10]. The proficiency of each laboratory to correctly identify the viruses in each pool was tested by allowing the laboratories to use their own bioinformatic pipelines. Seven different pipelines were used by the participants (two laboratories used the same pipeline; Table 2). Although the raw data contained all virus reads in each pool, not all laboratories correctly identified all the viruses. In particular, a new virus (AaNV), whose sequence was not available on GenBank at the time of the analyses, was only detected by three out of the eight laboratories. The reasons why five laboratories may have missed this virus are due to (a) a lack of plant virology expertise to interpret the data, (b) incorrect filtering or terms used to search the reference databases, and/or (c) rushed analysis.

Interestingly, false-positive results were also reported. AvCoV (*Coronaviridae*: *Coronavirinae*: *Gammacoronavirus*) is the causative agent of infectious bronchitis in several avian species, leading to multisystemic disease and economic loss in the poultry industry [13]. The AvCoV sequences were only reported by one laboratory (TPS_02, which outsourced the analysis) and could not be found in any other data set. It is likely that AvCoV reads found in the data of TPS_02 are due to a contamination from another sample in the same run on the sequencing platform or from the laboratory in which the extraction and ribodepletion were done (although this laboratory works with plant pathogens only). Contamination with reads from parallel samples in the same run was also observed in a previous study [14]. Although seven laboratories used the same sequencing provider, the samples were not sequenced in the same run, which might explain the presence of AvCoV reads in the dataset of TPS_02. The identification of the citrus exocortis Yucatan viroid by the same participant was based on NCBI Blast search. This false-positive finding is explained by the fact that the citrus exocortis Yucatan viroid sequence available on GenBank was found to be wrongly annotated and that it represents a mitochondrial RNA sequence instead [15,16]. There is a necessity for refining the databases used or building a new database with only verified sequences. This highlights the importance of using a reliable database. Under diagnostic settings, the expertise of the data analyst will have to be used to interpret whether these sequence reads should be regarded as a “true” diagnosis. In addition, virus host-range and symptoms should be taken into consideration when checking the virus findings. The presence of an unknown/new virus could be confirmed by an independent method e.g., RT-PCR.

Using the Virtool PathoscopeBowtie workflow, we failed to identify both PhCMoV and AaNV as they were not included in the plant virus OTUs database. An OTU typically represents a single virus species with one or more isolates attached to it and requires regular updates. However, using the NuVs workflow, we were able to identify both viruses after BLASTn search on the NCBI, thus confirming that all viruses were present in the raw data of all the participating laboratories. NuVs eliminates OTU viral reads, assembles contigs, and then predicts open reading frames (ORFs) in the contigs. The translated ORFs are scanned for viral protein motifs. The contigs can then be used for BLAST search directly on the NCBI and the results require a virology-expert interpretation. In the Virtool analysis, the viruses’ weight or the calculated proportion of reads mapping to each virus were low, which can be roughly proportional to low titre. In this case, the low weights reported in the analysis may not be attributed to the virus titre but to the reference database, as the majority of the OTUs used in this version of Virtool were not closely related to our viruses and only a few virus-isolate sequences were used by Virtool as references to establish the weight values. AV-1, CNDV, CLRV, ClYVV, and LNYV sequences generated in this study were very divergent isolates that shared low nucleotide identities to the most similar sequences in GenBank (Table 4). This demonstrates that Virtool-PathoscopeBowtie+NuVs is a promising bioinformatic tool that could be used for standardised bioinformatic analyses of HTS data for plant virus diagnostic setting.

One major advantage of Virtool is the use of curated databases that are checked for correct entries, thus limiting the potential of misidentifying viral reads. This means that the wrong detection of the citrus exocortis Yucatan viroid could have been avoided using the Virtool approach. Currently, a new version of Virtool is available with an updated OTUs database. However, this advantage is also a disadvantage as manpower is needed to update the databases regularly to include new virus and viroid sequences. This requires not only bioinformatics expertise but also expertise in virology.

The percentages of reads assigned to each virus varied among the different labs. Nevertheless, there were high percentages for the tobamoviruses, i.e., CGMMV and TMV, followed by CMV, a cucumovirus (Appendix A). This is also shown by the weights of the Virtool analysis (Figure 2). This might be explained by the strong silencing suppressors encoded by tobamoviruses and cucumoviruses, which will result in high virus titre in the infected plants [17,18,19]. The variation among the virus reads in the data provided by the participants could also be related to library preparations. The quantity and quality of the RNA provided for library preparation can affect the cDNA synthesis. Nevertheless, this did not affect the interpretation of the data and the detection of the viruses in the study.

HTS is a powerful tool for diagnostics that can identify unknown pathogens. As it can be seen from our study, bioinformatics expertise is required for the correct interpretation and diagnostics. Virtool provides a powerful open-source bioinformatics platform that can be implemented in a diagnostic laboratory. In addition, the data analysis with Virtool is reasonably straightforward in comparison to other bioinformatics approaches. Even though HTS costs have dropped, implementation of HTS for routine diagnostics is still cost-prohibitive, thus this method will probably be reserved for critical samples rather than routine diagnostics. It is worth noting that often the customer has no influence on the turnaround time at the sequencing supplier, thus this time needs to be taken into consideration. Currently, novel long-read sequencing technologies such as MinION from Oxford Nanopore Technologies may change this as the sequencing can be carried out relatively inexpensively in-house [20,21].

Diagnostic tests require sufficient controls in order to minimise the risk of false-negative or false-positive results [22]. These controls may include negative and positive extraction controls, negative and positive amplification controls, and internal propagation controls [22]. Internal propagation controls are usually targeting genomic sequences of the host matrix, such as *cox* or *nad5* [23,24]. However, for detection and the characterisation of plant viruses by HTS, these targets are not very practical. It may be therefore more useful to spike the plant material before extraction with a known amount of a plant virus that is not associated with this host. One example of a “positive virus control” is Phaseolus vulgaris endornavirus 1 and 2 (PvEV-1 and PvEV-2)-infected plant material; efficiency of the process from RNA extraction to HTS can be assessed by the recovery of viral reads of this virus [12,25]. It is also possible to add an extra control sample spiked with artificial RNAs of known sequences such as ERCC (external RNA controls consortium) [26]. This is particularly useful for monitoring the risk of cross-contamination on the sequencing platform if used in high concentrations.

## 4. Materials and Methods

### 4.1. Organisation

The study was organised by the Julius Kühn Institute (JKI; Germany) and assisted by the Netherlands Food and Consumer Product Safety Authority (NVWA).

### 4.2. Preparation of Virus-Infected Plant Material

*Nicotiana benthamiana* plants were inoculated mechanically with 14 different plant virus species (Table 5) as described in [27]. For each virus, infected plant leaves were homogenised in a Norit inoculation buffer (50 mM phosphate-buffer, pH 7 containing 1 mM of ethylenediaminetetraacetic acid [Na-EDTA], 20 mM of sodium diethyldithiocarbamic acid [Na-DIECA], 5 mM of thioglycolic acid, 0.75% activated charcoal, and 30 mg of Celite). The homogenate of each virus was gently rubbed onto individual healthy *N. benthamiana* leaves using a glass spatula. The inoculated plants were incubated under greenhouse conditions (22 °C; photoperiod of 16 h light and 8 h dark). Fifteen days post-inoculation, the non-inoculated leaves were tested using ELISA, PCR, or RT-PCR to confirm virus infection (Table 5). To ensure homogeneity, infected plant leaves were collected, ground, and freeze-dried using a Christ LCG LYO Chamber Guard, 121,550 PMMA, ALPHA 1-4 LSC (Martin Christ Gefriertrocknungsanlagen GmbH, Osterode am Harz, Germany). Freeze-dried samples were retested again for the presence of the viruses after storage for 2 years at 4 °C and were confirmed positive.

### 4.3. Chemicals Aliquoting and Shipment

Each participant was supplied with a sufficient aliquot of the RNeasy Plant Mini Kit (Qiagen) in order to carry out totRNA extraction according to the manufacturer’s instructions. In addition, aliquots of the RiboMinus™ Plant Kit for the RNA-Seq (Invitrogen) kit were also provided to each participant. The chemicals were shipped by fast courier (cool pack) and arrived at their destination within three days (except for one lab, Lab ID: TPS_07, for which shipment took 13 days).

### 4.4. The Sequential Steps That the Participants Were Requested to Perform

1.Preparation of pooled samples:

The lyophilised plant material for the HTS_TPS was mixed into two pooled samples (Pool 1 and Pool 2) and each laboratory was asked to combine the vials as in Table 3.

2.Total RNA extraction:

The RNeasy Plant Mini Kit was used for the isolation of total RNA from the pooled samples following Appendix A. The concentrations of the RNA extracts were measured using a Nanodrop spectrophotometer (Thermo Scientific, Waltham, MA, USA).

3.Ribosomal RNA depletion:

The RiboMinus™ Plant Kit for the RNA-Seq kit was used for depleting the ribosomal RNA following Appendix A.

4.High-throughput sequencing:

Ribodepleted RNA was sent for library construction (a customised library protocol based on the NEBNext Ultra II Directional DNA Library Prep Kit for Illumina with TruSeq Adapter sequences was used) and HTS was performed on the Illumina HiSeq 4000 platform (2 × 150 bp) at Eurofins Genomics GmbH, except for one laboratory in which the pools were sequenced at GenomeScan (Leiden, Netherlands) on the Illumina NextSeq 500 platform after library preparation using the NEBNext Ultra II Directional RNA Library Prep Kit for Illumina (New England Biolabs).

5.In-house bioinformatic analysis:

Each laboratory used their own bioinformatic pipeline to analyse the HTS data (Table 2).

6.Reporting the results:

A form was provided to each lab for collecting information on their procedures and data analysis, and for reporting the viruses detected.

7.Assessing Virtool:

Raw reads were uploaded to a JKI cloud server by each laboratory and then analysed on a single Virtool pipeline at JKI.

The above steps of the interlaboratory comparison are summarised in Figure 3.

### 4.5. Virtool Analysis

Each sample was analysed for viruses using Virtool (v3.9.8), which is a sample manager that can run multiple diagnostic analysis workflows [30]. For rapid identification of known viruses, the HTS pooled samples were aligned against the plant virus database using the PathoscopeBowtie approach [30,31]. PathoScope and Bowtie2 2.2.3+ were used to align reads to a plant viruses database derived from GenBank [30,31,32]. To identify novel viruses, the NuVs approach in Virtool was used followed by direct BLASTn search on the NCBI. Briefly, the HTS sample reads were mapped to known virus and host sequences using Bowtie2 [30,32]. The unmapped reads were assembled into contigs using SPAdes 3.8+ and viral coding regions in the contigs were predicted using the vFAM resource and HMMER 3.1b2+ [30,33,34,35]. The host genome of *N. benthamiana* was used (File: Niben.genome.v1.0.1.scaffolds.nrcontigs.fasta; downloaded from ftp://ftp.solgenomics.net/genomes/Nicotiana_benthamiana/assemblies/ (accessed on 25 August 2020)).

To normalise the reads from each laboratory, three subsamples of each of the two million reads were randomly extracted from the raw data sets provided by each laboratory using Geneious Prime (version 2021.1.1). The subsamples were then reanalysed with Virtool.

### 4.6. Statistical Analysis

The statistical analysis of the generated raw data sets and the subsamples (percentages, means, and standard error of mean [SEM]) was performed using R version 4.0.3 [36]. The data were visualised by ggplot2 package in R [37].

## 5. Conclusions

This study showed that the participants’ technical (wet lab) performance is sufficient to detect viruses in infected plants. The ribodepletion approach for HTS was found to be adequate for the detection of the plant viruses used in this study (ssDNA and ssRNA positive and negative senses). This strategy provides a good tool for plant virus diagnostics but the bottleneck concerns bioinformatic competence. The interlaboratory comparison underscores the range of bioinformatic pipelines used by the different participants for data analysis and the level of expertise. The inability to correctly identify new viruses or viruses not found in databases suggests that plant virology expertise is required. Virtool also allowed for the identification of all viruses, specifically twelve viruses (plus a satellite RNA) with PathoscopeBowtie and two viruses with the NuVs approach. Thus, Virtool can offer an alternative open-source platform for plant virus detection. This interlaboratory comparison was a very good exercise as all laboratories could re-check their data on their own and identify why they deviated in their analyses from the “correct” findings. This helped to improve and optimise their bioinformatic pipelines for plant virus diagnostics.

## Figures and Tables

**Figure 1 pathogens-10-01174-f001:**
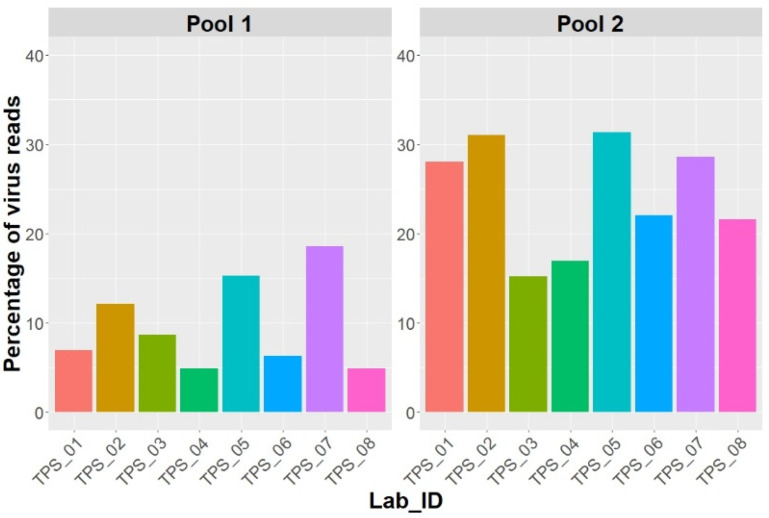
The percentage of reads mapped to viruses by the Virtool PathoscopeBowtie approach for the raw data provided by each participant.

**Figure 2 pathogens-10-01174-f002:**
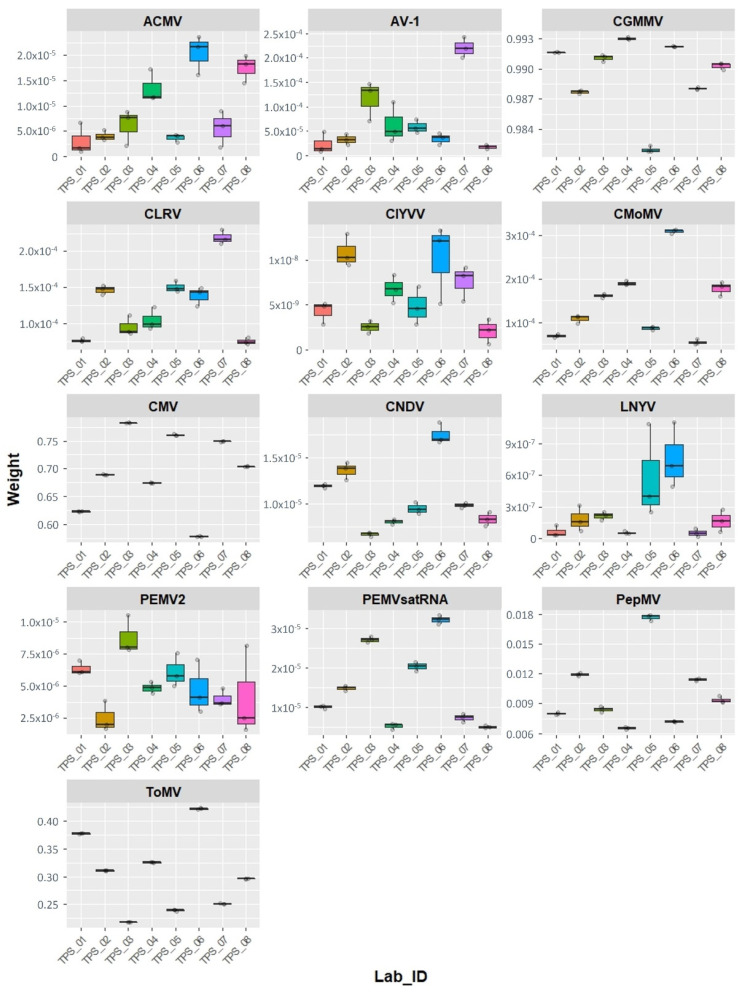
Box plot of the weights of viruses detected by the Virtool PathoscopeBowtie approach for the normalised subsets of two million reads (three replicates). The circles represent the replicates. The viruses are: ACMV, African cassava mosaic virus; AV-1, asparagus virus 1; CMoMV, carrot mottle mimic virus; CNDV, carrot necrotic dieback virus; CLRV, cherry leaf roll virus (strain carrot); ClYVV, clover yellow vein virus; CGMMV, cucumber green mottle mosaic virus; CMV, cucumber mosaic virus; LNYV, lettuce necrotic yellows virus; PEMV2, pea enation mosaic virus 2; PEMVsatRNA, pea enation mosaic virus satellite RNA; PepMV, pepino mosaic virus; and ToMV, tomato mosaic virus.

**Figure 3 pathogens-10-01174-f003:**
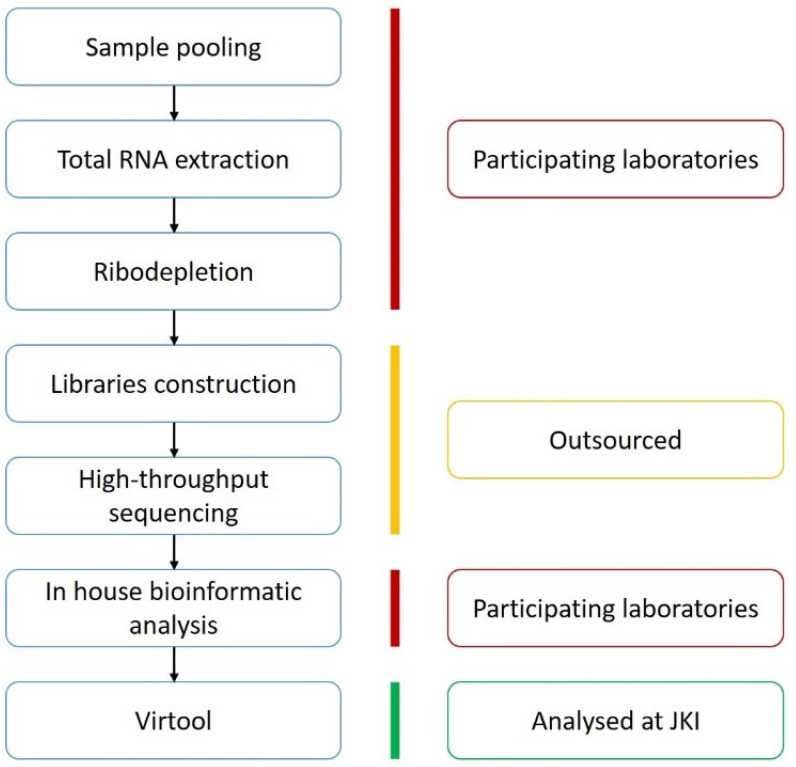
Graphical representation of the interlaboratory comparison. In red: work performed by the participating laboratories. In yellow: pooled samples sequenced at Eurofins Genomics GmbH, except for one laboratory which sequenced at Genomescan. In green: work done at the Julius Kuehn institute (JKI) cloud.

**Table 1 pathogens-10-01174-t001:** Characteristics of the raw data: the quantity of the totRNA and ribodepleted totRNA provided for sequencing, number of raw reads obtained, and the percentage of reads with a quality of (Q) ≥ 30. The raw data were reanalysed by Geneious for better comparison.

Lab ID	Pool 1				Pool 2			
	totRNA Quantity (ng/μL)	Ribodepleted totRNA Quantity (ng/μL)	Number of Raw Reads	Q ≥ 30	totRNA Quantity (ng/μL)	Ribodepleted totRNA Quantity (ng/μL)	Number of Raw Reads	Q ≥ 30
TPS_01	357.5	57.2	54,074,222	92.3	281.1	43.7	48,768,068	92.8
TPS_02	318.6	11.8	19,541,014	91.9	146	13.3	15,829,278	91.1
TPS_03	28.5	3.8	16,692,936	93.1	73	9.1	20,928,004	93.1
TPS_04	73.8	4.2	20,388,244	91.6	237.6	16.9	22,487,248	92.1
TPS_05 ^a^	73	5.7	75,831,846	83.2	86.8	4.2	79,993,710	84.1
TPS_06	245	21.8	39,932,780	93.5	249	25.5	35,501,376	94.1
TPS_07	Nr ^b^	405 ^c^	12,948,820	93.3	Nr ^b^	104 ^c^	9,871,160	92.6
TPS_08	20	7.7	17,740,834	91.9	225	24	19,539,256	91.2

^a^ Laboratory TPS_05 used a different sequencing provider. ^b^ Not recorded. ^c^ This unusual deviation could not be confirmed by participant number 7.

**Table 2 pathogens-10-01174-t002:** Bioinformatic tools and software used by the participating laboratories.

Lab ID	Bioinformatic Analysis	Complete Analysis Date
TPS_01	Geneious (Biomatters Limited): The raw reads were normalised using the *BBNorm* tool (version 38.84), quality trimmed, and size-filtered using Geneious. De novo assembly using the Geneious assembler (medium sensitivity/fast), followed by BLASTn search against virus/viroid sequences in the nt database on the NCBI non-redundant database (December 2018), was performed.	15 December 2018
TPS_02	SGA, IDBA_UD, SSPACE, Bowtie2, bbmap, Biopython, and BLASTn.	18 March 2019
TPS_03	CLC Genomic Workbench 11 custom-built pipeline (databases: NCBI viral RefSeq, November 2018; Pfam [v31]), DIAMOND (database: NCBI *nr*, June 2018).	31 December 2018
TPS_04	Cutadapt v1.16, Prinseq v0.20.4, PEAR v0.9.8, SortmeRNA v2.0, VirusDetect v1.5 with databank v229, CLC Genomic Workbench 10, and BLASTn/BLASTx.	13 November 2018
TPS_05	CLC Genomic Workbench 11, BLASTn, DIAMOND, and Krona (database September-2018).	8 December 2018
TPS_06	FastQC, Rcorrector, trim_galore, Bowtie2, Trinity, Samtools, and R packages: Iranges, dplyr, and BLASTn.	8 March 2019
TPS_07	VirusDetect v1.7 (Linux version) and BLASTn/BLASTx (database December 2018)	17 August 2019
TPS_08	Geneious (Biomatters Limited): The raw reads were normalised using the *BBNorm* tool (version 38.84), quality trimmed, and size-filtered using Geneious. De novo assembly using the Geneious assembler (medium sensitivity/fast), followed by BLASTn search against virus/viroid sequences in the nt database on the NCBI non-redundant database (October 2018), was performed.	30 November 2018

**Table 3 pathogens-10-01174-t003:** Virus sequences detected by the participating laboratories. The viruses are: ACMV, African cassava mosaic virus; AaNV, alfalfa-associated nucleorhabdovirus; AV-1, asparagus virus 1; CMoMV, carrot mottle mimic virus; CNDV, carrot necrotic dieback virus; CLRV, cherry leaf roll virus (strain carrot); ClYVV, clover yellow vein virus; CGMMV, cucumber green mottle mosaic virus; CMV, cucumber mosaic virus; LNYV, lettuce necrotic yellows virus; PEMV2, pea enation mosaic virus 2; PEMVsatRNA, pea enation mosaic virus satellite RNA; PepMV, pepino mosaic virus; PhCMoV, Physostegia chlorotic mottle virus; and ToMV, tomato mosaic virus.

Pool		Pool 1											Pool 2					
Virus		ACMV	AV-1	CGMMV	CLRV	ClYVV	CMoMV	LNYV	PEMV2	PEMVsatRNA	PepMV	Detection %	AaNV ^a^	CMV	CNDV	PhCMoV	ToMV	Detection Percentage
Lab ID	TPS_01	+	+	+	+	+	+	+	+	+	+	100	+	+	+	+	+	100
	TPS_02	+	+	+	+	−	+	+	−	−	+	70	−	+	−	+	+	60
	TPS_03	+	+	+	+	+	+	+	+	+	+	100	−	+	+	+	+	80
	TPS_04	+	+	+	+	+	+	+	+	−	+	90	−	+	+	+	+	80
	TPS_05	+	+	+	+	+	+	+	+	+	+	100	−	+	+	+	+	80
	TPS_06	+	+	+	+	+	+	+	+	−	+	90	+ ^b^	+	+	+	+	100
	TPS_07	+	+	+	+	+	+	+	+	−	−	80	−	+	−	+ ^c^	+	60
	TPS_08	+	+	+	+	+	+	+	+	+	+	100	+	+	+	+	+	100

^a^ AaNV was not available in the NCBI database at the time of the analysis by most laboratories. ^b^ This laboratory analysed the sample after the sequence of AaNV was available in the NCBI database. ^c^ This was reported as a new virus.

**Table 4 pathogens-10-01174-t004:** List of assembled virus sequences (accession numbers and their closest homologue GenBank accession). The raw data were reanalysed by Geneious to generate consensus sequences. The viruses are: ACMV, African cassava mosaic virus; AaNV, alfalfa-associated nucleorhabdovirus; AV-1, asparagus virus 1; CMoMV, carrot mottle mimic virus; CNDV, carrot necrotic dieback virus; CLRV, cherry leaf roll virus (strain carrot); ClYVV, clover yellow vein virus; CGMMV, cucumber green mottle mosaic virus; CMV, cucumber mosaic virus; LNYV, lettuce necrotic yellows virus; PEMV2, pea enation mosaic virus 2; PEMVsatRNA, pea enation mosaic virus satellite RNA; PepMV, pepino mosaic virus; PhCMoV, Physostegia chlorotic mottle virus; and ToMV, tomato mosaic virus.

Pool	Virus	Accession Number	Closest nt Sequence
Pool 1	ACMV	MW848516 and MW848517	97.3% and 95.4% X17095 and X17096
	AV-1	MW848534	90.4% KJ830761
	CGMMV	MW848531	99.7% MH271420
	CLRV	MW848518 and MW848519	85.1% and 82.9% KC937022 and FR851462
	ClYVV	MW848532	88.1% MN399730
	CMoMV	MW848525	97.4% FJ188472
	LNYV	MW848533	80.8% AJ867584
	PEMV2	MW848526	98.4% MN399707
	PEMVsatRNA	MW848527	95.9% U03564
	PepMV	MW848530	99.7% AJ606361
Pool 2	AaNV	MW848524	99.9% MG948563
	CMV	MW848520, MW848521, and MW848522	99.9%, 99.9%, and 99.7% D00356, D00355, and D10538
	CNDV	MW848523	76.9% NC_038320
	PhCMoV	MW848528	97.5% KY706238
	ToMV	MW848529	99.9% KY912162

**Table 5 pathogens-10-01174-t005:** Information list about the viruses used in the study. The list contains the virus confirmation methods (ELISA, PCR, and RT-PCR), including the antibodies and primers used, and the assigned pool for sequencing. Abbreviations: ssDNA, single-stranded DNA; (−ve) ssRNA, negative sense single-stranded RNA; and (+ve) ssRNA, positive sense single-stranded RNA.

Number	Virus	Acronym	Genus	Family	Genome		Confirmation	Primers/antibodies	Pool
1	African cassava mosaic virus	ACMV	*Begomovirus*	*Geminiviridae*	ssDNA	Bipartite	PCR	[28]	1
2	Alfalfa-associated nucleorhabdovirus ^a^	AaNV	*Betanucleorhabdovirus*	*Rhabdoviridae*	(−ve) ssRNA	Monopartite	ELISA	JKI-1607	2
3	Asparagus virus 1	AV-1	*Potyvirus*	*Potyviridae*	(+ve) ssRNA	Monopartite	ELISA	JKI-44	1
4	Carrot mottle mimic virus	CMoMV	*Umbravirus*	*Tombusviridae*	(+ve) ssRNA	Monopartite	RT-PCR	JKI-881 TACCCTAACATGTACGCCGC and JKI-882 GCGTTCAGATATTGCCGCTG	1
5	Carrot necrotic dieback virus	CNDV	*Sequivirus*	*Secoviridae*	(+ve) ssRNA	Monopartite	ELISA	JKI-45	2
6	Cherry leaf roll virus (strain carrot)	CLRV	*Nepovirus*	*Secoviridae*	(+ve) ssRNA	Bipartite	ELISA	[29]	1
7	Clover yellow vein virus	ClYVV	*Potyvirus*	*Potyviridae*	(+ve) ssRNA	Monopartite	ELISA	JKI-98	1
8	Cucumber green mottle mosaic virus	CGMMV	*Tobamovirus*	*Virgaviridae*	(+ve) ssRNA	Monopartite	ELISA	JKI-1773	1
9	Cucumber mosaic virus	CMV	*Cucumovirus*	*Bromoviridae*	(+ve) ssRNA	Tripartite	ELISA	JKI-1745	2
10	Lettuce necrotic yellows virus	LNYV	*Cytorhabdovirus*	*Rhabdoviridae*	(−ve) ssRNA	Monopartite	ELISA	JKI-2073	1
11	Pea enation mosaic virus 2 ^b^	PEMV2	*Umbravirus*	*Tombusviridae*	(+ve) ssRNA	Monopartite	RT-PCR	[12]	1
12	Pepino mosaic virus	PepMV	*Potexvirus*	*Alphaflexiviridae*	(+ve) ssRNA	Monopartite	ELISA	JKI-1452	1
13	Physostegia chlorotic mottle virus	PhCMoV	*Alphanucleorhabdovirus*	*Rhabdoviridae*	(−ve) ssRNA	Monopartite	ELISA	JKI-2051	2
14	Tomato mosaic virus	ToMV	*Tobamovirus*	*Virgaviridae*	(+ve) ssRNA	Monopartite	ELISA	JKI-68	2

^a^ Novel virus at the time of the study. ^b^ Associated with pea enation mosaic virus satellite.

## Data Availability

Not applicable.

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
