# Peer review of "Interlaboratory Comparison Study on Ribodepleted Total RNA High-Throughput Sequencing for Plant Virus Diagnostics and Bioinformatic Competence"

_pathogens, 2021, doi:10.3390/pathogens10091174_

Round 1

Reviewer 1 Report

Gaafar et al. performed an inter-laboratory comparison study to evaluate the use of ribodepleted total RNA in the standard implementation in HTS for plant virus diagnostics.  HTS technologies have been used widely in the recent years to detect and to characterize novel plant viruses.  Considering the great potential of the HTS technologies in the future, the information provided in this study is extremely useful and this manuscript should be published. Having said that, the study was designed with a few flaws that may be related to cost-prohibitive facts and some bias that may have been out of their hands. The main flaws I found were: RNA extraction from pooled freeze-dried samples, rather than pooled samples prepared from equimolar RNA from individual samples, and the extreme variability in the number of reads obtained in HTS even though the same company was used for HTS for 7 out of the 8 samples. This last flaw may have been facility-related rather than anything else. The authors' findings are very interesting (accurate and precise diagnosis was based on expertise and bioinformatic approaches used), but they must be more properly discussed for which I have added my comments and suggestions. Also, authors should be careful about the proper reference of tables and figures in the main text. The manuscript shall be accepted, only if they properly implement the revisions. Finally, I also provided some feedback/corrections on some parts so that the language of the manuscript can be improved.

Author Response

Reviewer 1:

Gaafar et al. performed an inter-laboratory comparison study to evaluate the use of ribodepleted total RNA in the standard implementation in HTS for plant virus diagnostics.  HTS technologies have been used widely in the recent years to detect and to characterize novel plant viruses.  Considering the great potential of the HTS technologies in the future, the information provided in this study is extremely useful and this manuscript should be published. Having said that, the study was designed with a few flaws that may be related to cost-prohibitive facts and some bias that may have been out of their hands. The main flaws I found were: RNA extraction from pooled freeze-dried samples, rather than pooled samples prepared from equimolar RNA from individual samples, and the extreme variability in the number of reads obtained in HTS even though the same company was used for HTS for 7 out of the 8 samples. This last flaw may have been facility-related rather than anything else. The authors' findings are very interesting (accurate and precise diagnosis was based on expertise and bioinformatic approaches used), but they must be more properly discussed for which I have added my comments and suggestions. Also, authors should be careful about the proper reference of tables and figures in the main text. The manuscript shall be accepted, only if they properly implement the revisions. Finally, I also provided some feedback/corrections on some parts so that the language of the manuscript can be improved.

L30: Change from 'ribosomal RNA depletion approach' to 'ribosomal RNA depleted total RNA'. The latter is wider used in the literature.

Edited.

L35: Please, mention what kind of contaminations? Cross-contamination in the lab?

Cross-contaminations at the lab or at sequencing provider. Edited.

L40: Change from 'ribodepletion enrichment' to 'ribodepletion target enrichment'. You are enriching the transcriptome target by ribodepleting, which is the aim in transcriptome and virus discovery studies.

Edited.

L68-71: Please, add one or two sentences that compare (ds)RNA enrichment method with sRNA and ribodepleted totRNA. In our experience, HTS of dsRNA have allowed the full or partial characterization of dsDNA, (+)ss-RNA, (-)ss-RNA as well as dsRNA viruses.

Added to the discussion.

L73: change from 'validate the approach' to 'validate an standard approach'

Edited.

L74: change from 'plant viruses detection' to 'plant virus detection'

Edited.

L77: Suggestion: change from 'Another challenge for' to 'Perhaps the most critical challenge for'

Edited.

L92: Delete the parentheses between 'which' and '(PT)'.

Edited.

L98: You mean table 1?

Edited.

L101: The minimum value, I see in table1 in ribodepleted totRNA quantity, for pool 1 is 3.8 ng/uL (not 3.75), while the maximum is 57.2 ng/uL (not 30). Similarly, the maximum value for pool 2 is 43.7 ng/uL (not 28.5). The outlier values that I see for both pool 1 and 2 come TPS_07, not from TPS_06.

Edited.

L103-105: Out of curiosity, why is there so much variation in the number of reads for the different pooled samples that were sequenced. From 13-54 M of reads and from 10-49 M of reads is so much variation in the output reads. It would have been ideal that this number be kept as homogenous as possible considering the influence of the number of reads (deep of HTS) in virus discovery and diagnostics studies. In my experience in US-based HTS facilities, the client can ask for a specific number of HTS reads. I wonder if this is not the same in Europe.

The sequencing provider Eurofins Genomics was asked for 5 million reads x2 (1.5 GB). Nevertheless, we received different amounts of reads. It is common for us to get more reads than what we asked for. In case of TPS_01, the asked for 10 million reads x2. I do not know if this is my case. In case of Genomescan, was asked for 2 Gbase output.

L105: RNA integrity (RIN) values are not detailed in table 1, therefore add after 'generated raw reads': '(data not shown)'.

Added.

L111: Change from 'SRR14794355 to SRR14794340' to 'SRR14794340 to SRR14794355'

Edited.

L112: Table 1. How was ribodepleted totRNA amount inferred by each laboratory, nanopore or qubit or what? This is not described in materials and methods. Please, add this information. Also, consider changing the numbers representation from '54074222' to '54'074,222' which helps to ease reading.

Nanodrop. Edited.

Furthermore, consider adding another column in table 1 that details the amount of totRNA (before ribodepletion) to provide an idea how much RNA is 'sufficient RNA for further processing' which is stated in line 98.

Added.

L113-114. 'The raw data ... better comparison' Please briefly specify what was done with Geneious? read normalization?

At this point, the reason for using Geneious, that one lab reported exceptionally high Q30 and Q20 using their bioinformatics pipeline. Thus, the consortium decided to use one software to check the quality of the reads.

L115, change to 'used a different'

Edited.

L119, change 'where they outsourced' to 'that outsourced'

Edited.

L125: citrus exocortis Yucatan viroid does not exist, therefore it should not have an acronym. As you discuss later, citrus exocortis Yucatan viroid is an incorrectly annotated mitochondrial sequence. I would suggest to use the whole word throughout the manuscript to stress this is not a viroid.

Edited.

Table 2, top align TPS_01 through TPS_08 so that they exactly match to the beginning of the information for the bioinformatic analysis used.

Edited.

Table 3, suggestion: add another column as 'Date of analysis' which will display the date when the bioinformatic analysis was performed. This will be particularly useful considering the apparent lack of availability of both AaNV of PhCMoV in the viral databases of Genbank when some analyses were performed.

Added.

L139: This was reported as a new virus by TPS_07? Please, rephrase

Edited.

L142, change 'form' to 'from'

Edited.

L143, I wonder what was the divergence/similarity among the consensus sequences obtained from each dataset, at the intraspecies level? Add a brief sentence detailing this information.

This was not analysed.

Table 4, change 'closest nt sequences on GenBank' to 'closest homolog GenBank accession'

Edited.

L167-169, this information should be added to one table or figure, maybe figure 1, per pooled sample, per virus, or at least per virus.

Added.

L168, operational taxonomic unit (OTU)?

We used the term  as defined  by the virtool developers.

L175-176, Change from 'The same was observed' to 'A similar scenario was observed'

Edited.

Figure 1, suggestion: if graphically attractive, consider representing all this data as box plots which also shows the repetitions, instead of just the repetitions.  One boxplot per TPS. This should be doable in the powerful R with ggplot.

Edited.

Discussion needs considerable work.

- Add some discussion about the limitation of your variation in the number of reads obtained per TPS. Even though you have variation in the number of reads, still all the viruses were detected. However, the variation of number of reads must be mentioned because the deep of HTS (number of reads) can also lead to false-positives.

Added.

- Add possible explanations of why some plant viruses were found more abundantly than others in the HTS bioinformatic analysis?

Added.

- Furthermore, mention that even though your study used ribodepleted totRNA to successfully detect ssDNA and ssRNA plant viruses, other studies have also used ribodepleted totRNA for diagnosing and characterizing dsDNA and dsRNA plant viruses.

Added.

- Discuss briefly about the preference in study of preparing the pooled samples by pooling the freeze-dried material rather than pooling equimolar RNA from individual samples. The fact that pooled freeze-dried samples were used may insert some bias for the RNA preps to contain more nucleic material of some samples over others (it is very hard to completely homogeneize freeze-dried materials). I understand this may have been done due to costs, still this must be mentioned.

Added.

- Finally, discuss about the different pipelines used by each TPS and the possible explanation on why some TPS failed to detect/diagnose some plant viruses. How I see it, this is very likely to the fact that some TPS tied their analysis to nucleotide-level reference-based comparisons, while protein-level non-reference based comparisons are needed for the best detection and characterization of new viruses or divergent viral strains (BLASTx and virus protein domains identification in the ORFs).

This is not the case as the three laboratory who identified the new virus sequence only using blastn.

L200: After 'lead to false-negative results.', add something like 'if the HTS is not deep enough (more number of reads are needed), which makes HTS more cost-prohibitive'.

Edited.

L201: Ribodepletion reduces the amount of host and microbiome-associated host rRNA.

Edited.

L206: change to Illumina HiSeq4000 machine and add the Illumina or whatever HTS platform was used in Genomescan

Edited.

L208: change to correctly identify

Edited.

L209-210: in this sentence add the reference to the table that details the bioinformatic pipelines used by each TPS.

Edited.

L211: change to correctly identified

Edited.

L212-213: change 'that had not been published' to 'that was not available'

Edited.

L214: you can not guarantee these were reasons, change to 'may have missed'

Edited.

L225-227: where you able to find AvCoV sequences when you re-analyzed the raw data from TPS_02??

Yes.

L227 and 228: do not use the acronym CEYVd for citrus exocortis Yucatan viroid as it is not a viroid.

Edited.

L228: change 'This is an error as the CEYVd sequence listed in the NCBI datase' to 'This false-positive is explained by the fact that the citrus exocortis Yucatan viroid sequence available on Genbank'

Edited.

L230: Rather than reference number 13, a more proper citation that clarifies that citrus exocortis Yucatan viroid is not a viroid is: https://doi.org/10.1007/s00705-014-2200-6

Added.

L235: after this sentence, add a brief sentence explaining what was the limit of your study: none of the HTS findings were validated by PCR because your study limited to the HTS and bioinformatic analyses. This must be said.

The viruses in the study were confirmed during the preparation of the samples and later to confirm again their presence in the freeze dried material after long time storage.

Additional sentence was added.

L241: what kind of BLAST search: BLASTn, BLASTx, tblastx?

BLASTn, Edited.

L242: change 'viral reads and assemble contigs then predict' to 'viral reads, assemble contigs and then predict'

Edited.

L245: requires a virology expert interpretation??

Edited.

L245' Rephrase this sentence 'Moreover ... to low titre'

Edited.

L246: the virus read count in a HTS data is not only biased by the bioinformatic analyses, but mostly by the library preparation method (cDNA synthesis bias and more importantly the PCR bias).

The same library preparation was used at Eurofins but on different batches. Only for TPS_05 NEBNext Ultra II Directional RNA Library Prep Kit was used by the different sequencing provider.

Edited.

L249: sequences generated in this study

Edited.

L249: change 'as they' to 'that'

Edited.

L249: mention a range of the nucleotide identity level these viruses showed in the BLAST search.

Table 4, Edited.

L250: change to 'Virtool-PathoscopeBowtie+NuVs'

Edited.

L256: rephrase this sentence 'However, ... as manpower is needed'

L260: change 'as it' to 'that'

Edited.

L261: change to 'as it can be seen'

Edited.

L261: change 'identification' to 'interpretation and diagnostics'.

Edited.

L262-264: rephrase this sentence: 'Virtool offers .... other bioinformatic approaches.' Suggestion: do not use relatively easy, because what is relatively easy for some, can be extremely challenging for others.

Edited.

L264-265: change 'However ... are still very high' to 'Even though HTS costs have drop, implementation of HTS for routine diagnostics is still cost-prohibitive'

Edited.

From L260-271, the authors have used however three times. They need to re-organize these ideas to provide the advantages of Virtool and the bioinformatic approaches used in this study and then contrast with the drawbacks.

Edited.

L269-270: change 'such as the Oxford ... as MinION' to 'such as MinION from Oxford Nanopore Technologies'

Edited.

L272-281: Rephrase this whole paragraph. It is hard to completely understand what authors intend to mean.

Edited.

L288: Do you mean table 5? Please correct.

Edited.

L289: change from '50mM pH 7 phosphate-buffer' to '50 mM phosphate-buffer, pH 7'

Edited.

L294: please specify if local or systemic leaves were tested.

Non-inoculated leaves, edited.

L295: Table 5, not table1

Edited.

L298: Freeze-dried samples were retested again, using the same detection method?. Please clarify

Yes.

L307: Change from 'QIAGEN' to 'Qiagen'

Edited.

L313: Change from 'Participants were requested to perform' to 'The sequential steps that the participants were requested to perform were: '

Edited.

L313: Consider adding a new subsection 4.4 for the HTS-TPS. Then, Virtool analysis would be subsection 4.5

Edited.

L324: Please, provide a more detailed library preparation kit used (brand). This information to be displayed is essential for harmonized standarized HTS study. Different library preps and brands can provide different results mainly due to different steps involved in the library prep.

Added.

What Illumina or HTS platform was used at Genomescan? If it was Illumina, please specify what Illumina platform was used.

Illumina Nextseq 500 platform. Added.

L331-332: Please, rephrase this sentence.

Edited.

L334: it would be important to be briefly mentioned how the raw data were shared considering the raw data file size in HTS studies and your specific purpose of TPS in HTS

The files were sent online to JKI Cloud and then used of analysis. Edited.

L355-357: Move this paragraph to L344 after 'diagnostic analysis workflows [22].' to favor easiness of reading

In the first part, the data were analysed in total, but the second paragraph was for comparison.

L363: Define what is sufficient. To me the fact that some TPS failed to find/diagnose some viruses is not sufficient. Talking about regulatory samples with quarantine pathogens, accurateness and precision are a must.

We agree with the reviewer, but here we consider the wet lab procedures that were enough to enrich and find all the viruses in the study in all raw data sets, thus the wet lab procedures are sufficient.

L367 change 'put emphasis on' to 'underscore'

Edited.

L369-371. Please rephrase this sentence: 'The bioinformatic ... easily missed'

Edited.

L377: change to 'bioinformatic pipelines for plant virus diagnostics'

Edited.

Reviewer 2:

HTS has become an essential tool in virus diagnostics. The extreme sensitivity of the sequencing methods also comes with issues of false positives/ negatives if proper methods are not followed even for those who use it routinely. It is therefore very important to identify the potential pitfalls in the post sequencing analysis methodologies that are used in diagnosing the viruses present in a specific sample.  The current study was designed and conducted to address some of these challenges. There is a need for more of similar kind of studies to standardize the sequence analysis methods. Even though the conclusions from the current study are nothing novel but overall the study was conducted methodically and the results are presented clearly. I recommend the manuscript for publication

-L30: "To assess if HTS of ribosomal RNA depletion approach is "fit for purpose" as diagnostic tool" You are not assessing  whether ribosomal RNA depletion is useful or not. You keep stressing on the ribodepletion at several places. Your study is not comparing ribosomal RNA depletion vs. non-depletion. That is just start material and everyone knows the advantages of this approach. You need not to stress on it putting more emphasis on current experimental methods

Edited.

Reviewer 2 Report

HTS has become an essential tool in virus diagnostics. The extreme sensitivity of the sequencing methods also comes with issues of false positives/ negatives if proper methods are not followed even for those who use it routinely. It is therefore very important to identify the potential pitfalls in the post sequencing analysis methodologies that are used in diagnosing the viruses present in a specific sample.  The current study was designed and conducted to address some of these challenges. There is a need for more of similar kind of studies to standardize the sequence analysis methods. Even though the conclusions from the current study are nothing novel but overall the study was conducted methodically and the results are presented clearly. I recommend the manuscript for publication

-L30: "To assess if HTS of ribosomal RNA depletion approach is "fit for purpose" as diagnostic tool" You are not assessing  whether ribosomal RNA depletion is useful or not. You keep stressing on the ribodepletion at several places. Your study is not comparing ribosomal RNA depletion vs. non-depletion. That is just start material and everyone knows the advantages of this approach. You need not to stress on it putting more emphasis on current experimental methods

Author Response

(The authors gave the same response as above.)
